# Assessment of healthcare waste management practices and associated factors in Addis Ababa City Administration Public Health Facilities

Menelik Legesse Tadesse[1], Bethabile Lovely Dolamo[2]*

1 Menelik II Medical and Health Science College, Addis Ababa, Ethiopia, 2 University of South Africa, Pretoria, South Africa

* dolamobethabilelovely@gmail.com

## Abstract

**Data Availability Statement:** All manuscript files are available from the UNISA database URI: http://hdl.handle.net/10500/26614.

### Background

Healthcare waste management is very important due to its hazardous nature that can cause risk to human health and the environment. In developing countries, healthcare waste has not received much attention and has been disposed of together with municipal waste. The aim of the study was to assess the healthcare waste management practices in Addis Ababa City Administration Public Health Facilities.

### Methods

An institutional-based cross-sectional design was used for the study at Addis Ababa city 15 Public health centres and 3 hospitals. Data were collected using self-administered questionnaires distributed to 636 randomly selected healthcare waste handlers and managers. Observational check list also used. The data were entered into the EPI- INFO version TM 7 and exported to IBM SPSS 20 for analysis. Both descriptive and analytic statistics were employed.

### Results

Among the respondents, 358 (90.86%) from health centres and 96.38% (133) from hospitals indicated that their facilities had separate containers for hazardous and non-hazardous waste however, 61 (15.48%) from health centres and 29 (21.01%) from hospitals indicated that healthcare waste containers were not clearly marked or labelled. The study found that the main forms of on-site treatment of healthcare waste for health centres and hospitals before disposal were burning. Manager respondents from the health centres 65 (92.86%), 64 (91.43%) and from hospitals 31 (91.18%), 30 (88.24%) indicated that healthcare waste handlers were used protective clothing when handling waste and were provided with protective clothing when handling healthcare waste respectively.

**Funding:** The author(s) received no specific funding for this work.

**Competing interests:** The authors have declared that no competing interests exist.

## Conclusion

In this study healthcare waste management among healthcare waste handlers and healthcare facility managers were not getting full attention. Collection of healthcare waste were not done regularly, containers were not clearly marked and were not located in appropriate areas where they might be needed. Support healthcare waste handlers by training help to improve their knowledge, attitude and practice.

## Background

Healthcare waste produced in the course of health care activities entails a higher risk of infection and injuries than municipal waste. In developing countries, healthcare waste has not received much attention and has been disposed of together with municipal waste [1]. In Ethiopia, improper healthcare waste management is alarming and poses a serious threat to public health [2].

The risk of healthcare waste and its management has become a global cause of concern. The majority of the problems are associated with an exponential growth in the health care sector together with low or non-compliance with guidelines and recommendations. The management of healthcare waste requires increased attention and diligence to avoid substantial disease burden associated with poor practice, including exposure to infectious agents and toxic substances [3, 4]. According to the United Nations Environmental Program [5], healthcare waste is one of the most troublesome forms of waste and one of the most important environmental concerns for the global community. Healthcare waste production at hospitals and its management are important issues worldwide [6]. Since the mid-1990s the world has experienced a dramatic increase in the amount of hazardous waste generated, at the same time, a vigorous drive for sustainable development and increased environmental awareness and concern [7].

Poor waste management practices at the level of healthcare facilities, including failure to segregation of waste and errors in the colour coding of waste disposal, can result in hazardous waste being disposed of not only improperly, but also accessible to community members [8]. In Botswana, [9] found that due to a lack of understanding of the importance of colour coding and segregation in the management of healthcare waste, patients were given healthcare waste bags for their personal belongings and clothing after being discharged from the hospital. In Korea [10] found that policy on healthcare waste management was inadequate and required strengthening.

The poor management of healthcare waste (HCW) is associated with a lack of adequate training of healthcare workers and disposal practices, including disposal with municipal waste together with some autoclave treatment and incinerator use.

The studies conducted in Ethiopia health centres and hospitals focused on healthcare waste generation and did not consider its management and intervention [2, 11]. The high generation of healthcare waste is due to the increasing population and the use of healthcare facilities that exceeds the ability of Addis Ababa City Administration to manage the increased amount of healthcare waste. This study wished to assess the management system The concern is about the lack of appropriate HCW segregation, selection, handling, storage, transport, treatment and final disposal. This motivated the researcher to conduct this study to assess the healthcare waste management in health facilities in Addis Ababa City Health Bureau. Between 2011 and 2016, the Addis Ababa City Administration Health Bureau built more than 60 health centres and one [1] referral hospital [12].

## Materials and methods

### Study setting and design

The study setting was Addis Ababa, the capital city of Ethiopia. It is the largest and most populous city in the country [13]. The city has three layers of administration, the city administration at the top, 10 sub-cities administration in the middle and 116 Woredas (Districts) at the bottom [12]. There were 6 public referral hospitals and 95 functional public health centers during the study period. A facility based cross-sectional study was conducted among healthcare waste handlers and managers. The study assessed the healthcare waste management practices in 15 selected public health centres and 3 hospitals. Data was obtained from questionnaires distributed to 636 randomly selected healthcare professionals, ancillary staff and managers from January 24 to February 24, 2018.

### Source population

All healthcare waste handlers and managers in Addis Ababa City Administration public health centres and hospitals.

### Study population

Sampled healthcare waste handlers and managers from selected health centres and hospitals.

### Inclusion and exclusion criteria

Healthcare waste handlers and managers (Doctors, Health Officers, Nurses, midwifes, pharmacists, laboratory technicians, Environmental health professionals, Biomedical engineers, ancillary staffs comprised cleaners, porters and operatives for handling waste selected by proportion.) in 15 health centres and 3 hospitals who were worked more than six months and agree to participate in the study were included. Healthcare waste handlers and managers who were absent at the time of data collection were excluded from the study.

### Sample size and sampling techniques

The sample size determined for this study was determined by a single population proportion formula with the assumption of 50%, 95% confidence interval and 5% marginal error. The researcher used multistage sampling in this study and calculate the design effect of 2. Correction formula was also used because the number of healthcare workers were less than 10000. A total of 636 participants were selected out of which 532 were healthcare waste handlers and 104 were managers. Proportional allocation was performed 394 participant healthcare waste handlers were from 15 health centres and 138 were from 3 hospitals. Moreover 70 managers were from health centres and 34 were from hospitals. Simple random sampling method was used to select participants from both health centres and hospitals.

### Data collection tool

Data was collected by means of questionnaires and observational check list. To reduce subjectivity (information bias), the principal investigator adopted a structured questionnaire from World Health Organization's healthcare waste management rapid assessment tools [14] as a data collection tool in line with the research objective. The questionnaire included respondents' demographic characteristics, knowledge and practice of HCW management. The questionnaire consisted of closed questions (requiring a 'Yes' or 'No' answer). The main questions covered segregation, collection, transportation, storage, treatment and disposal, waste

recycling and re-use, occupational health and safety, internal policies, and administration and healthcare waste management. Data collectors distributed the questionnaires in the 15 health centres and 3 hospitals to the respondents. Observation was conducted by the data collectors and supervisors to see the waste management practice of healthcare waste handlers and the work site to health centres and hospitals. The data collectors used the prepared observational check list to follow the HCW management practice and captured supporting photographs.

## Data quality control

Fifteen data collectors who graduated from a college with Grade 10+3 diploma in health science were used for data collection on healthcare waste management. Eight supervisors who were BSc graduates in Environmental Health or related fields assisted the principal investigator with the data collection. Training manual was prepared for two days of training. The principal investigator gave training to data collectors and supervisors, including data collection and fieldworkers practice in data collection and data-collection tools. Information sheet and consent form also attached with the data collection tool to share the information about the study. Data collection tools and observational check lists was pre-tested to two health centers and one hospital other than the study areas. Respondents who were not able to read, the English language questionnaires were the tool were translated to local language Amharic by professional translator and assisted by the data collectors. Daily onsite supervision was made by the supervisors and principal investigator during data collection assuring ethical issue and respondents assuring an animosity.

## Data processing and analysis

Data was entered by EPI- INFO TM 7 after a manual check for completeness. The entered data were exported to IBM SPSS Version 20. Both descriptive and inferential statistics were used. Data analysis was performed separately for each of health centres and hospitals which were grouped by category. Tables and graphs were used to show frequencies, percentage, bivariate logistic regression analysis to identify candidate variables for multivariable logistic regression analysis. The multivariable analysis a significant association was found with a p-value of less than 0.05. the association were presented with an adjusted odds ratio (AOR) and corresponding 95% CI.

## Ethical consideration

Ethical approval and clearance were obtained from the Higher Degrees Committee, Department of Health Studies, University of South Africa and Addis Ababa City Administration Health Bureau to conduct the study. The letter was submitted to both health centers and hospital administrators to begin the study. A written information sheet and consent form was provided to all participants. The participants were informed of the purpose of the study; that participation was voluntary, and that all information would be treated strictly confidentially. The participants signed informed consent and also informed to withdraw from the study at any time was clearly stated for the participants.

# Results

## Sociodemographic characteristics of the respondents

A total of 636 healthcare waste handlers and managers, 370 (58.18%) and 266 (41.82%) were males and females participated respectively (Table 1). In this study 15 health centres and 3 general hospitals were selected (Figs 1 and 2). The largest 251 (39%) participants were nurses 162

**Table 1. Respondents' gender and distribution at the study healthcare facilities, Addis Ababa City Administration, February 2018.**

| Gender | Hospital | Health centre | Total | Percent |
|--------|----------|---------------|-------|---------|
| Male | 72 | 194 | 266 | 41.82 |
| Female | 100 | 270 | 370 | 58.18 |
| Total | 172 | 464 | 636 | 100.00 |

(64.5%) worked at health centres and 89 (35.5%) worked at hospitals, the least 2 (0.31%) participants were biomedical engineers worked at hospitals (Fig 3).

The respondents' age ranged from 20 to 59 years. Of the respondents, 372 (58.49%) were aged between 20–29, 216 (33.96%) were aged between 30–39, and 5.03% (32) were aged between 40–49. The mean age of the respondents was 30.9 years (Table 2).

Of the respondents, 421 (66.19%), 158 (24.84%), 24 (3.77%) and 21 (3.3%) had work experience of, 1–5 years, 6–10 year, 11–15 years and 21 years and more experience respectively (Table 3).

## Healthcare waste management practice

From this study healthcare waste handlers, 350 (88.83%) from health centres and 127 (92.03%) from hospitals indicated that the health facility they worked at had separate storage areas for HCW. With reference to storage, 358 (90.86%) from health centres and 133 (96.38%) from hospitals indicated that their facilities had separate containers for hazardous and non-hazardous waste. The respondent healthcare waste handlers, 61 (15.48%) from health centres and 29 (21.01%) from hospitals indicated that the healthcare waste containers were not clearly marked or labelled (Table 4).

Healthcare waste handlers, 339 (86.04%) from health centres and 106 (76.81%) from hospitals indicated that the HCW containers were located appropriate areas where they might be needed. Majority of respondent healthcare waste handlers, 325 (82.49%) and 83. 330 (76%)

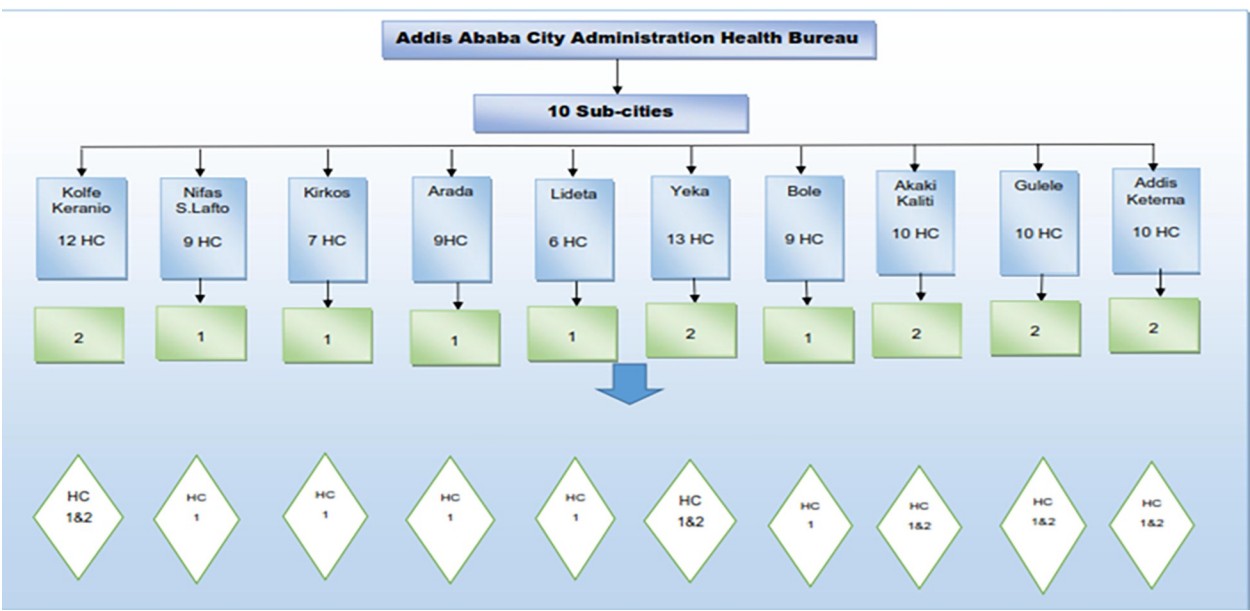

**Fig 1. Number of health centres selected from ten sub-cities of Addis Ababa City Administration, February 2018.**

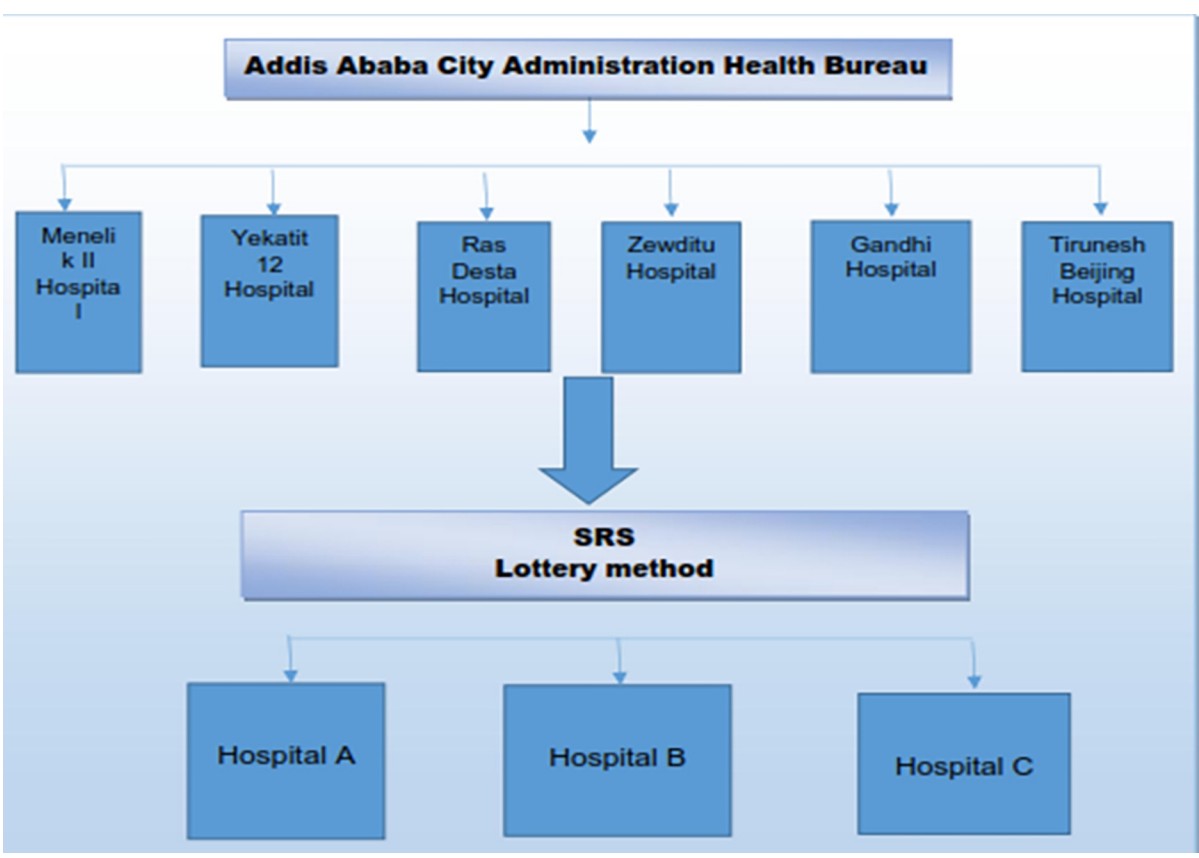

**Fig 2. Number of hospitals selected from Addis Ababa City Administration, February 2018.**

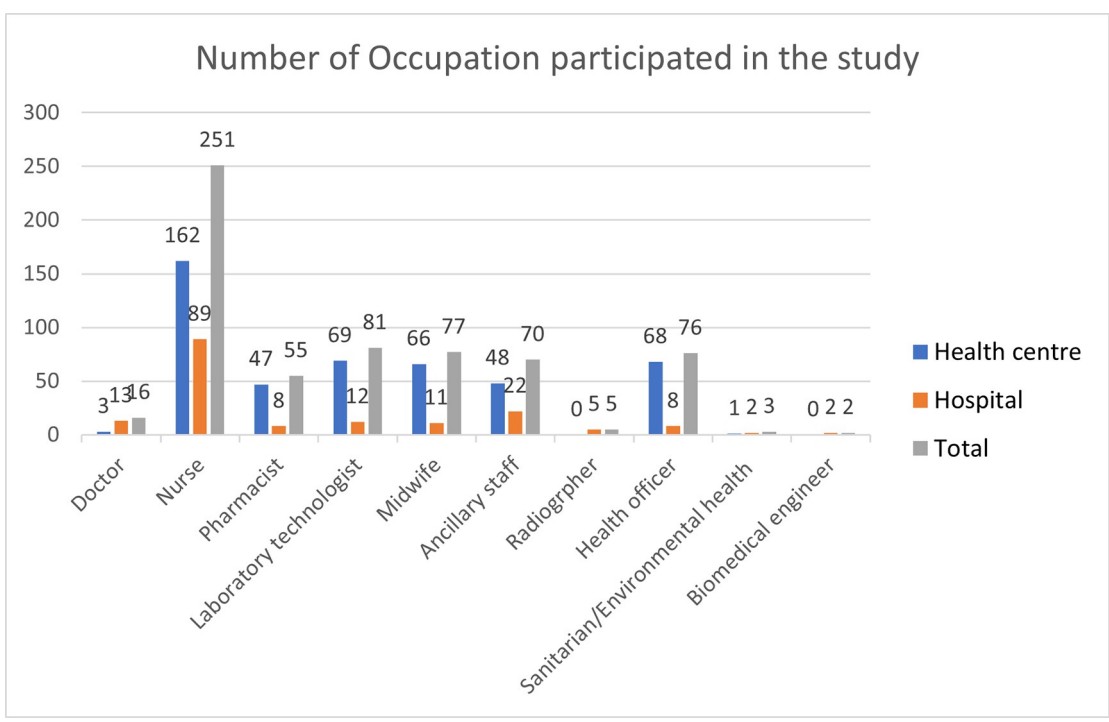

**Fig 3. Number of respondents by occupation, Addis Ababa City Administration, February 2018.**

**Table 2. Respondents' age distribution at the study healthcare facilities, Addis Ababa City Administration, February 2018.**

| Age group (N = 636) | Number of respondents | Percent |
|---|---|---|
| 20–29 | 372 | 58.49 |
| 30–39 | 216 | 33.96 |
| 40–49 | 32 | 5.03 |
| 50–59 | 16 | 2.52 |
| 60+ | 0 | 0.00 |
| Total | 636 | 100.00 |

from health centres and 105 (76.09%) and 99 (71.74%) from hospitals indicated that the HCW containers in health facilities were made of leak-proof material and the HCW containers were easy to carry respectively (Table 4).

Respondents from health centres343 (87.06%) and 119 (86.23%) from hospitals stated that the sharps containers were made of puncture-resistant material but 51 (12.94%) from health centres and 41 (22.71%) from hospitals indicated that sharps containers were not closed securely and disposed of whenever they were 3/4 full. Most of the respondents, 335 (85.02%) from health centres and 87 (63.04%) from hospitals stated that the HCW containers were emptied daily or whenever they were 3/4 full (Table 4).

Healthcare waste handlers from health centres 96 (24.37%) and 41 (29.71%) from hospitals indicated that no formal or informal separation of waste took place at their health facilities. Respondents from health centres 209 (53.03%) and 87 (63.04%) from hospitals indicated that plastics and intravenous sets were not kept separately for recycling. Healthcare waste handlers, 126 (31.98%) and 310 (76.68%) from health centres and 49 (35.51%) and 93 (67.39%) from hospitals indicated that not all waste handlers wore heavy duty gloves and sturdy shoes when handling HCW and washed their hands and their hard duty gloves after handling waste respectively (Table 4).

The respondent healthcare handlers 157 (39.85%), 260 (65.99%), 58.12% (229), 246 (62.44%) and 56 (14.21%) from health centres and 61 (44.2%), 97 (70.29%), 93 (67.39%), 90 (65.22%) and 38 (27.54%) from hospitals indicated cytotoxic, pathological, reagent, outdated pharmaceutical and radioactive waste was indicated their facility generated as waste of special concern (Table 4).

None of the respondents who indicated that their facilities generated HCW of special concern indicated how the disposal thereof was handled. The respondents were asked to indicate how liquid waste was disposed of and to specify for cytotoxic and reagent processing liquids. Healthcare waste handlers, 20 (5.08%) from health centres and 10 (7.25%) from hospitals

**Table 3. Study respondents' work experience at the study health facilities, Addis Ababa City Administration, February 2018.**

| Respondents' work experience | Number of respondents(n = 636) | Percent |
|---|---|---|
| 1–5 | 421 | 66.19 |
| 6–10 | 158 | 24.84 |
| 11–15 | 24 | 3.77 |
| 16–20 | 12 | 1.89 |
| 21+ | 21 | 3.30 |
| Total | 636 | 99.99 |

**Table 4. HCW handlers' management practice at the study health facilities, Addis Ababa City Administration, February 2018.**

| Questions | Health centre (n = 394) | | Hospital (n = 138) | | Total (n = 532) |
|---|---|---|---|---|---|
| | Yes | No | Yes | No | |
| Does the facility have a separate area or separate storage areas for HCW? | 350 (88.83%) | 44 (11.17%) | 127 (92.03%) | 11 (7.97%) | 532 |
| Does the facility have separate containers for non-hazardous and hazardous waste? | 358 (90.86%) | 36 (9.14%) | 133 (96.38%) | 5 (3.62%) | 532 |
| Are all types of waste containers clearly marked or labelled? | 333 (84.52%) | 61 (15.48%) | 109 (78.99%) | 29 (21.01%) | 532 |
| Are all types of containers located in appropriate areas where they might be needed? | 339 (86.04%) | 55 (13.96%) | 106 (76.81%) | 32 (23.19%) | 532 |
| Are containers made of leak-proof material (preferably plastic) for disposal of HCW? | 325 (82.49%) | 69 (17.51%) | 105 (76.09%) | 33 (23.91%) | 532 |
| Are the containers easy to carry by the workers? | 330 (83.76%) | 64 (16.24%) | 99 (71.74%) | 39 (28.26%) | 532 |
| Are sharps containers made of a puncture-resistant material (cardboard, plastic, or metal)? | 343 (87.06%) | 51 (12.94%) | 119 (86.23%) | 19 (13.77%) | 532 |
| Are HCW containers emptied daily or whenever they are 3/4 full? | 335 (85.02%) | 59 (14.97%) | 87 (63.04%) | 51 (36.96%) | 532 |
| Are sharps containers closed securely and disposed of whenever they are 3/4 full? | 343 (87.06%) | 51 (12.94%) | 97 (70.29%) | 41 (29.71%) | 532 |
| Does any formal or informal separation of waste take place? | 298 (75.63%) | 96 (24.37%) | 97 (70.29%) | 41 (29.71%) | 532 |
| Are used plastics and intravenous sets kept separately for recycling? | 185 (46.95%) | 209 (53.05%) | 51 (36.96%) | 87 (63.04%) | 532 |
| Do all waste handlers wear heavy duty gloves and sturdy shoes when handling medical waste? | 268 (68.02%) | 126 (31.98%) | 89 (64.49%) | 49 (35.51%) | 532 |
| Do waste handlers? wash their heavy-duty gloves and their hands after handling HCW? | 310 (76.68%) | 84 (21.32%) | 93 (67.39%) | 45 (32.61%) | 532 |
| Does the establishment generate any waste of special concern: | | | | | |
| • Cytotoxic? | 157 (39.85%) | 237 (60.15%) | 61 (44.20%) | 77 (55.8%) | 532 |
| • Pathological waste? | 260 (65.99%) | 134 (34.01%) | 97 (70.29%) | 41 (29.71%) | 532 |
| • Reagent? | 229 (58.12%) | 165 (41.89%) | 93 (67.39%) | 45 (32.61%) | 532 |
| • Out-dated pharmaceuticals? | 246 (62.44%) | 148 (37.56%) | 90 (65.22%) | 48 (34.78%) | 532 |
| • Radioactive waste? | 56 (14.21%) | 338 (85.79%) | 38 (27.54%) | 100 (72.46%) | 532 |
| If yes, how is their disposal handled? | - | - | - | - | - |
| How is liquid waste disposal? | Sinks 20 (5.08%) Sewer 23 (5.84%) | 43 (10.91%) | Sinks 10 (7.25%) Sewers 4 (2.90%) | 14 (10.14%) | 57 |

indicated that liquid waste was disposed of via sinks, and 23 (5.84%) from health centres and 4 (2.9%) from hospitals indicated via sewers. None of the respondents specified cytotoxic or reagent processing liquids (Table 4).

## Types of HCW generation

The types of HCW generated and observed in respective healthcare facilities in daily basis were asked to the study participants. Most respondents observed, 277 (70.30%), 261 (66.24%) and 265 (67.26%), 114 (82.61%) from health centres and hospitals indicated used gloves and sharps respectively were generated more in daily basis (Table 5).

The type of HCW least generated and observed from health centres and hospitals 80 (20.30%), 67 (48.55%) and 60 (15.23%), 59 (42.7%) were indicated human tissue and organ and excreta respectively (Table 5).

**Table 5. Types of HCW generated daily at the selected healthcare facilities, Addis Ababa City Administration, February 2018.**

| Type of healthcare waste generated and observed | Health centre (n = 394) | | Hospital (n = 138) | |
|---|---|---|---|---|
| | Number of respondents | Percent of respondents | Number of respondents | Percent of respondents |
| Dressing swabs, genital swabs, absorbents | 215 | 54.57 | 92 | 66.67 |
| Used sanitary pads | 130 | 33.0 | 59 | 42.75 |
| Used gloves | 277 | 70.30 | 118 | 85.51 |
| Fluids | 125 | 31.73 | 79 | 57.25 |
| Used bandages | 185 | 46.95 | 79 | 57.25 |
| Human tissue and organs | 80 | 20.30 | 67 | 48.55 |
| Excreta | 60 | 15.23 | 59 | 42.75 |
| Sharps (used cannulas, needles, surgical blades, vials injections, syringes) | 265 | 67.26 | 114 | 82.61 |
| General waste or non-infectious | 261 | 66.24 | 101 | 73.19 |
| Used toilet paper | 136 | 34.52 | 71 | 51.45 |

The respondents were asked to indicate the on-site means of transportation observed of HCW in their healthcare facilities. The study found that health centres mainly used buckets followed by pedal bins and trolleys to transport HCW on site while hospitals used mainly pedal bins and sometimes buckets and trolleys (Fig 4).

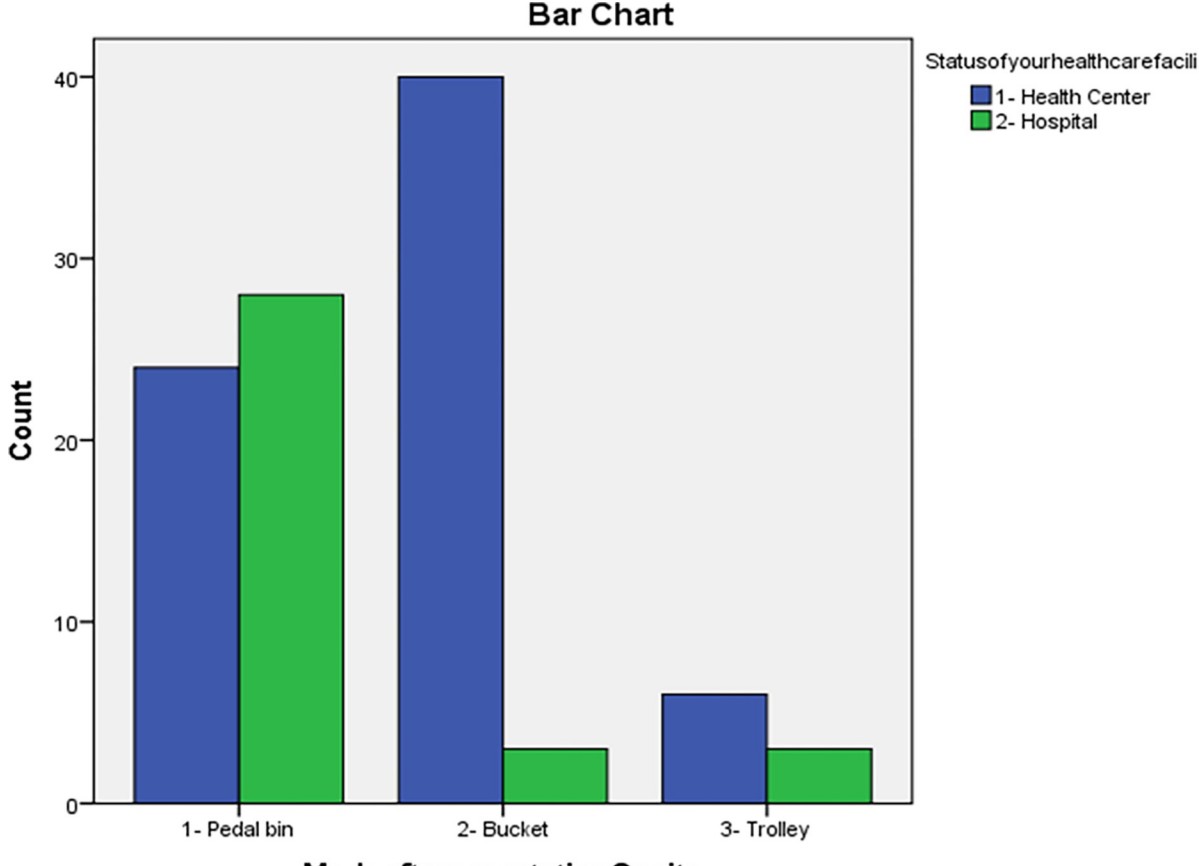

**Fig 4. On-site means of transportation for HCW in the selected healthcare facilities, Addis Ababa City Administration, February 2018.**

**Table 6. Number of managers profession participated in the study healthcare facilities, Addis Ababa City Administration, February 2018.**

| Profession | Frequency | Percent |
|---|---|---|
| Nurse | 37 | 35.6 |
| Pharmacist | 12 | 11.5 |
| Laboratory technologist | 15 | 14.4 |
| Midwife | 12 | 11.5 |
| Ancillary staff | 9 | 8.7 |
| Radiographers | 2 | 1.9 |
| Health officers | 15 | 14.4 |
| Sanitarian/Environmental health | 2 | 1.9 |

## Healthcare waste management and managers

Regarding HCW management issues, a total of 104 managers respondents, 70 from health centres and 34 from hospitals were participated. The type of professions to managers participated in the study were nurses, laboratory technicians and health officers accounted 37 (35.6%) and 15 (14.4%) each respectively (Table 6).

Manager respondents, 69 (98.57%) from health centres and 31 (91.18%) from hospitals indicated that HCW generated by their facilities was segregated. From health centres 47 (61.43%) and from hospitals 24 (70.59%) indicated that the HCW was securely stored before transportation to the incinerator. Healthcare manager from the health centres 65 (92.86%) and 31 (91.18%) from hospitals indicated that healthcare waste handlers used protective clothing when handling waste, and 64 (91.43%) from health centres and 30 (88.24%) from hospitals indicated that the waste handlers were provided with protective clothing when handling HCW (Table 7).

Managers from health centres 47(67.14%) and hospitals 21(61.76%) indicated that there were a current operational standard for HCW management in their health facilities. Forty-nine (70.0%) from health centres and 25(73.53%) from hospitals indicated they had applicable guideline for HCW in the health facilities, beside 57(81.43%) and 23(67.65%) from health centres and hospitals respectively indicated HCW management committee organized in healthcare facilities (Table 7).

The respondent managers were asked to indicate the type of protective clothing used for handling HCW in health centres, 45 (64.29%) used gloves; 35 (50%) used gowns; 17 (24.29%)

**Table 7. Managers response on HCW management in the study health facilities, Addis Ababa City Administration, February 2018.**

| Questions | Health centre (n = 70) | | Hospital (n = 34) | | Total (n = 104) |
|---|---|---|---|---|---|
| | Yes | No | Yes | No | |
| Is healthcare waste generated in your healthcare facility segregated? | 69 (98.57%) | 1 (1.43%) | 31 (91.18%) | 3 (8.82%) | 104 |
| How is healthcare waste awaiting transportation to the incinerator stored? | Secure 47 (61.43%) | Insecure 23 (38.57%) | Secure 24 (70.59%) | Insecure 10 (29.41%) | 104 |
| Do you the waste handlers use protective clothing when handling healthcare waste? | 65 (92.86%) | 5 (7.14%) | 31 (91.18%) | 3 (8.82%) | 104 |
| Do you provide the handlers/workers with protective clothing when handling healthcare waste? | 64 (91.43%) | 6 (8.57%) | 30 (88.24%) | 4 (11.76%) | 104 |
| Is there a current operational standard for HCW management? | 47 (67.14%) | 23 (32.86%) | 21 (61.76%) | 13 (38.23%) | 104 |
| Are there any applicable national, regional, and local guidelines for HCW management in the health centre? | 49 (70.0%) | 21 (30%) | 25 (73.53%) | 9 (26.47%) | 104 |
| Is there a healthcare waste management committee? | 57 (81.43%) | 13 (18.57%) | 23 (67.65%) | 11 (32.35%) | 104 |

**Table 8. Managers response to protective clothing used in the study health facilities, Addis Ababa City Administration, February 2018.**

| Type of protective clothing | Health centre (n = 70) | | Hospital (n = 34) | |
|---|---|---|---|---|
| | Number of respondents | Percent of respondents | Number of respondents | Percent of respondents |
| Gloves | 45 | 64.29 | 21 | 61.76 |
| Gowns | 35 | 50.00 | 9 | 26.47 |
| Aprons | 17 | 24.29 | 11 | 32.35 |
| Sturdy shoes | 23 | 32.86 | 9 | 26.47 |
| Goggles | 10 | 14.29 | 12 | 35.29 |
| Caps | 5 | 7.14 | 3 | 8.82 |
| Mask | 20 | 28.57 | 8 | 23.53 |

used aprons; 23 (32.86%) used sturdy shoes; 10 (14.29%) used goggles; 5 (7.14%) used capes, and 20 (28.57%) used masks. Of the respondents from hospitals, 21 (61.76%) used gloves; 9 (26.47%) used gowns; 11 (32.35%) used aprons; 9 (26.47%) used sturdy shoes; 12 (35.29%) used goggles; 3 (8.82%) used capes, and 8 (23.53%) used masks (Table 8).

The respondent managers were asked to rate their facilities' handling and segregation of HCW in health centres, 37 (52.86%) rated the handling of HCW good; 20 (28.57%) rated it very good; 7 (10%) rated it excellent, and 6 (8.57%) rated it poor. The respondent managers in the hospitals, 16 (47.06%) rated the handling good; 13 (38.24%) very good; 5 (14.71%) poor, and none rated it excellent (Fig 5A).

Respondent managers in health centres, 34 (48.57%) rated the segregation good; 23 (32.86%) very good; 8 (11.43%) poor, and 5 (7.14%) excellent. Manager respondents from hospitals, 18 (52.94%) rated the segregation good; 10 (29.41%) rated it very good; 5 (14.71%) rated it poor, and 1 (2.94%) rated it excellent (Fig 5B).

The manager respondents were asked to indicate the method and means of collection and off-site disposal of HCW, 14 (20%) from health centres and 7 (20.60%) from hospitals indicated that the municipality collected the HCW for off-site disposal. Of the respondents, 1 (1.43%) from the health centres and 1 (2.94%) from the hospitals indicated that a cooperative

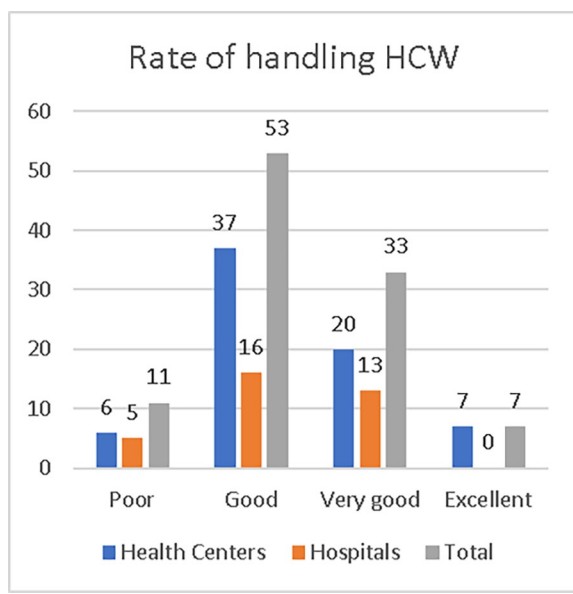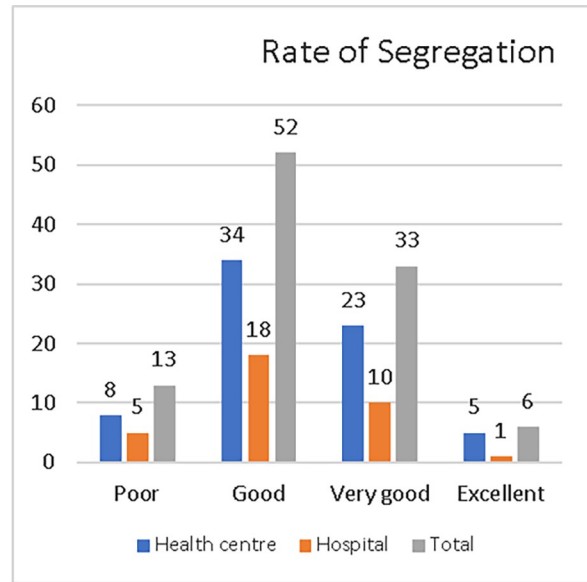

**Fig 5.** Rate of handling (a) and segregation (b) of HCW in the study health facilities, Addis Ababa City Administration, February 2018.

**Table 9. Managers response to off-site HCW collectors in the study health facilities, Addis Ababa City Administration, February 2018.**

| Type of organisation | Health centre (n = 70) | | Hospital (n = 34) | |
|---|---|---|---|---|
| | Number of respondents | Percent of respondents | Number of respondents | Percent of respondents |
| Municipality | 14 | 20.00 | 7 | 20.60 |
| Cooperative organization | 1 | 1.43 | 1 | 2.94 |

The frequency of off-site HCW collection from health centres 19 (27.14%) and hospitals 8 (23.53%) indicated in daily basis; 8.57% (6) from health centres and 9 (26.47%) from the hospitals indicated once a week, and 1 (1.43%) from the health centres indicated once a fortnight (Table 10).

organisation was responsible for collection and off-site HCW disposal (Table 9). Most of the respondents did not answer the question.

The manager respondents were asked what was used to store hazardous HCW prior to disposal, from health centres, 34 (48.57%) indicated red plastic healthcare waste bags; 16 (22.86%) indicated yellow sharps containers; 11 (15.71%) indicated 'other' and specified large interim containers; 5 (7.14%) indicated black plastic refuse bags; 3 (4.29%) indicated pedal bins; 1 (1.43%) indicated standard metal dustbins. From hospitals, 58.82% [20] indicated red plastic healthcare waste bags; 6 (17.65%) yellow sharps containers; 4 (11.76%) pedal bins; 2 (5.88%) black plastic refuse bags; 1 (2.94%) indicated standard metal dustbins, and 1 (2.94%) indicated 'other' and specified large interim containers (Fig 6).

The respondent managers were asked how HCW was transported on-site for storage before collection for off-site disposal, from health centres, 24 (34.29%) indicated in pedal bins; 40 (57.14%) indicated buckets, and 6 (8.57%) indicated trolleys. Respondent managers from hospitals, 28 (82.35%) indicated pedal bins; 3 (8.82%) indicated buckets, and 3 (8.82%) indicated trolleys (Table 11).

Managers were asked who was responsible for HCW management in their facilities, in health centres, 28 (40%) indicated sanitarian/environmental health professionals were responsible for HCW management; 27 (38.57%) indicated safety officers, and 15 (21.43%) indicated 'other' and specified (laboratory technicians, midwifes, ancillary staffs) (Fig 7).

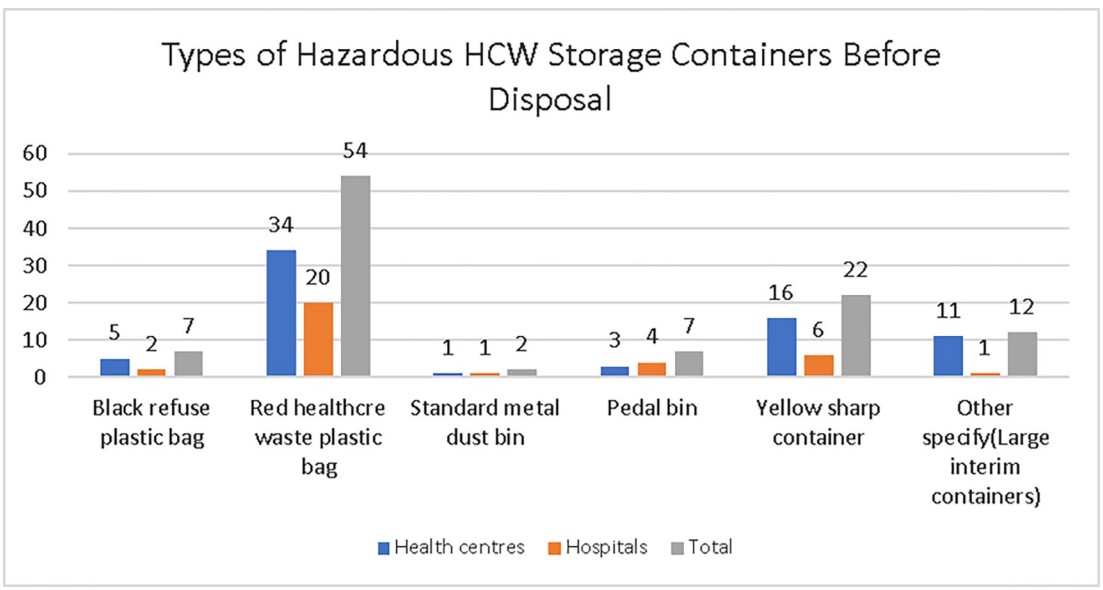

**Fig 6. Type of Hazardous HCW storage containers before disposal in the study health facilities, Addis Ababa City Administration, February 2018.**

**Table 10. Managers response to frequency of HCW collection by off-site authorities in the study health facilities, Addis Ababa City Administration, February 2018.**

| Time of HCW collection (off-site) | Health centre (n = 70) | | Hospital (n = 34) | |
|---|---|---|---|---|
| | Number of respondents | Percent of respondents | Number of respondents | Percent of respondents |
| Daily | 19 | 27.14 | 8 | 23.53 |
| Once a week | 6 | 8.57 | 9 | 26.47 |
| Once per fortnight | 1 | 1.43 | - | - |

**Table 11. Managers response to mode of on-site transportation used in the study health facilities, Addis Ababa City Administration, February 2018.**

| On-site HCW mode of transportation | Health centre (n = 70) | | Hospital (n = 34) | |
|---|---|---|---|---|
| | Number of respondents | Percent of respondents | Number of respondents | Percent of respondents |
| Pedal bin | 24 | 34.29 | 28 | 82.35 |
| Bucket | 40 | 57.14 | 3 | 8.82 |
| Trolley | 6 | 8.57 | 3 | 8.82 |

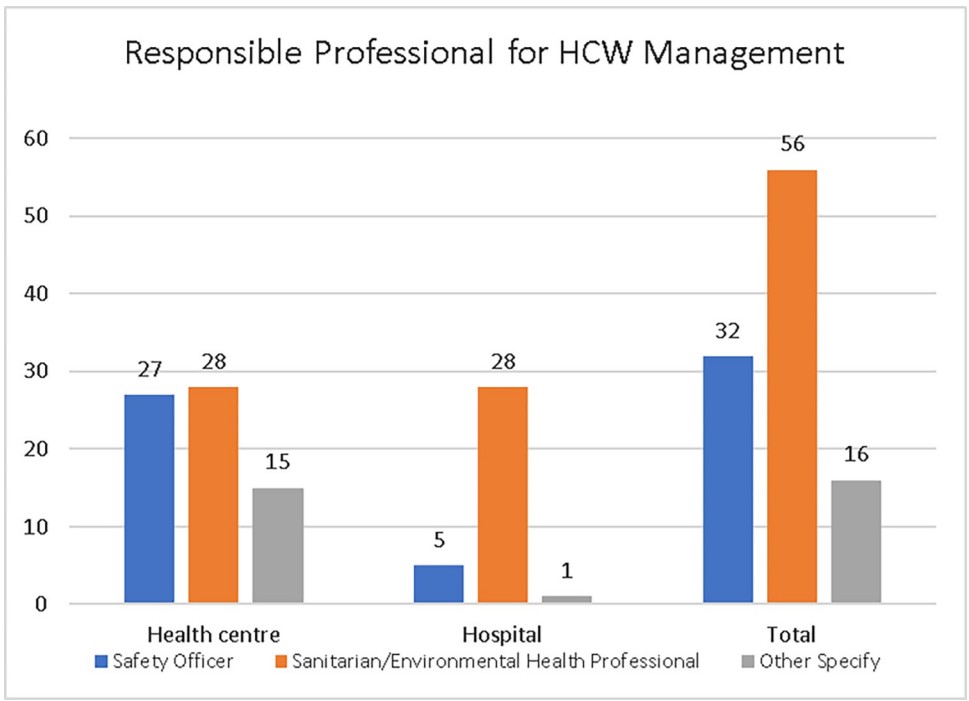

**Fig 7. Responsibility for HCW management in the study health facilities, Addis Ababa City Administration, February 2018.**

Managers from hospitals, 28 (82.35%) indicated sanitarian/environmental health professionals were responsible for HCW management; 5 (14.71%) indicated safety officers, and 1 (2.94%) indicated 'other' and specified ancillary staff (Fig 7).

## Risks of the current waste management system

The respondent managers were asked to indicate whether their health facilities had concerns about HCW management, 64 (91.43%) from health centres and 30 (88.24%) from hospitals

**Table 12. Managers concerns about HCW management in the study health facilities, Addis Ababa City Administration, February 2018.**

| Questions | Health centre (n = 70) | | Hospital (n = 34) | | Total (n = 104) |
|---|---|---|---|---|---|
| | Yes | No | Yes | No | |
| Does the management of the health facility have concerns about HCW management? | 64 (91.43%) | 6 (8.57%) | 30 (88.24%) | 4 (11.76%) | 104 |
| Does the waste pose any risk to waste collectors? If yes, what kind? | 35 (50%) | 35 (50%) | 21 (61.76%) | 13 (38.24%) | 104 |
| Was anyone getting injured by needles in the past 12 months and reported? | 19 (27.14%) | 51 (72.86%) | 24 (70.59%) | 10 (29.41%) | 104 |
| Does the health facility have registration book/a register for any injury or HCW contamination to the collectors/handlers? | 40 (57.14%) | 30 (42.86%) | 24 (70.59%) | 10 (29.41%) | 104 |

indicated that management had concerns about HCW management. Managers from health centres 35 (50.0%) and from hospitals 21 (61.76%) indicated that the HCW posed risks to waste collectors; 19 (27.14%) from health centres and 24 (70.59%) from hospitals indicated that waste collectors (handlers) had been injured by needles. Respondent managers, 40 (57.14%) from health centres and 24 (70.59%) from the hospitals indicated that their facilities had a register for injury or HCW contamination to staff (Table 12).

The respondent managers were asked to indicate the number of HCW handlers (ancillary/janitors) working at their facilities, from health centres, 59 (84.29%) indicated 5 or more; 3 (4.29%) indicated 4; 5 (7.14%) indicated 3, and 1 (1.43%) indicated 1. Manager respondents from hospitals, 29 (85.29%) indicated 5 or more; 3 (8.82%) indicated 2, and 2 (5.88%) indicated 1 (Fig 8).

The Manager respondents were asked to indicate the type of injuries sustained in their health facilities in the previous 12 months, from health centres, 8 (11.43%) indicated deep injuries; 10 (14.29%) indicated slight skin penetration; 5 (7.14%) indicated superficial, and 7 (10%) indicated splash injuries. From hospital managers, 12 (35.29%) indicated deep injuries; 15 (44.12%) indicated slight skin penetration; 14 (41.18%) indicated superficial, and 13 (38.24%) indicated splash injuries (Table 13).

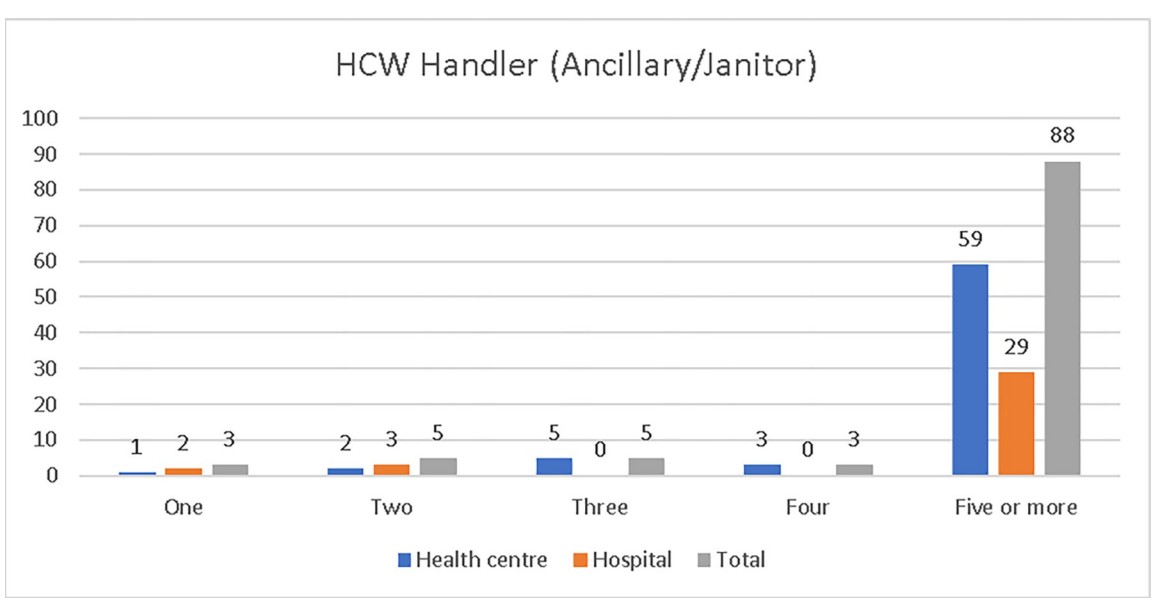

**Fig 8. Number of HCW handlers (ancillary/Janitors) in the study health facilities, Addis Ababa City Administration, February 2018.**

**Table 13. Managers response to types of injury sustained in the study health facilities, Addis Ababa City Administration, February 2018.**

| Type of injury | Health centre (n = 70) | | Hospital (n = 34) | |
|---|---|---|---|---|
| | Number of respondents | Percent of respondents | Number of respondents | Percent of respondents |
| Deep injury | 8 | 11.43 | 12 | 35.29 |
| Slight skin penetration | 10 | 14.29 | 15 | 44.12 |
| Superficial | 5 | 7.14 | 14 | 41.18 |
| Splash | 7 | 10.00 | 13 | 38.24 |

## Observation

The tide line of waste management with reference to waste minimisation, segregation, storage, handling, collection, and treatment was not properly and adequately practised by any of the surveyed health centres and hospitals. During the study 15 health centres and 3 hospitals selected were observed.

## Interim storage

Of the health care facilities, 13 health centres and 1 hospital had interim storage sites and HCW disposal sites located in areas minimally accessible to their staff. Six health centres and 2 hospitals had interim storage containers that had no lids to prevent odour and escape of wastes and waste leakage. Open plastic buckets and safety boxes were used to transport waste manually to the disposal site. In 10 health centres and the 3 hospitals HCW stored on site remained on site for more than 48 hours before final disposal (Table 14).

## Treatment and disposal of HCW

During the study period, almost all the health centres and hospitals did no use disinfection of HCW storage/collection utilities treatment (used chemical treatment or autoclaving) for HCW before disposal for on-site and off-site. Almost all the health centres and hospitals disposed of all HCW inside their compounds (on-site) as incineration considered the final treatment except placenta and surgically removed body parts. During observation, 1 health centre and 3 hospitals disposed of the HCW outside the compound (off-site) (Table 15). The disposals of ash residues were seen to the field near by the incinerator during observation (Fig 9). The burial site for placenta and surgical removals and were away from any water source at most of the health centres and hospitals. All the health facilities except 1 health centre had incinerators on the premises. In 3 of the health facilities, the incinerators were located downwind from the main service area. The incinerators of 11 health centres and 2 hospitals had sufficient air inlets on the side. At 12 of the health centres and all the 3 hospitals ash residues from the incinerators was disposed of inside the compound. The incinerators at 8 of the health centres and 2 of the hospitals were not surrounded by a fence or wall to limit access to scavengers

**Table 14. Interim storage for HCW observed in the study health facilities, Addis Ababa City Administration, February 2018.**

| Questions | Health centre (n = 15) | | Hospital (n = 3) | | Total (n = 18) |
|---|---|---|---|---|---|
| | Yes | No | Yes | No | |
| Are all interim storage sites and healthcare waste disposal sites located in areas that are minimally accessible to staff? | 13 | 2 | 1 | 2 | 18 |
| Do interim storage containers have lids? | 9 | 6 | 1 | 2 | 18 |
| Is waste stored on site for more than 48 hours before final disposal? | 5 | 10 | - | 3 | 18 |

**Table 15. Observed treatment and disposal of HCW in the study health facilities, Addis Ababa City Administration, February 2018.**

| Questions | Health centre | | Hospital | | Total |
|---|---|---|---|---|---|
| | Yes | No | Yes | No | |
| Is there any treatment of HCW before disposal? (if any chemical, autoclaving, crashing of needles) | 2 | 13 | - | 3 | 18 |
| If yes, how the residuals handled? If yes, how are the residuals handled? | Chemical disinfection | - | - | - | 2 |
| Is the health care waste disposed of | | | | | |
| On-site? | 15 | - | 3 | - | 18 |
| Off-site? | 1 | 14 | 3 | - | 18 |
| Is there an incinerator at healthcare facility? | 14 | 1 | 3 | - | 18 |
| Is the incinerator located downwind from the health centre? | 12 | 3 | 3 | - | 18 |
| Does the incinerator have sufficient air inlets on the side? | 11 | 4 | 2 | 1 | 18 |
| Where is the ash from the incinerator disposed of? | On-site 12 | Off-site 3 | On-site 3 | Off-site - | 18 |
| Is the incinerator surrounded by a fence or wall to limit access? | 7 | 8 | 1 | 2 | 18 |
| Is the burial site away from any water source at the healthcare facility? | 12 | 3 | 2 | 1 | 18 |
| Is the pit 1–2 meters wide and 2–5 meters deep? Is the bottom of the pit at least 1.8 meters above the water table? | 13 | 2 | 3 | - | 18 |
| What type of HCW is burned in the incinerator? | All types of HCW | | All types of HCW | | |

(Fig 10A and 10B). Burial pits such as placenta pits and surgical removal pits were employed for final on-site waste disposal. The burial pits in most of the health centres and hospitals was 1–2 meters wide and 2–5 meters deep and the bottom of the pit was at least 1.8 meters above the water table (Table 15).

During observation all healthcare facilities used incineration for on-site HCW disposal except 1 health centre. The health centre that did not incinerate HCW disposed of it by open burning in the premises (Table 16). All hospitals used municipality for off- site disposal moreover incineration for disposal of HCW. Most, 12 health centres had no off-site disposal for HCW some 3 of the health centres used cooperative organization for HCW off-site disposal beside incineration (Table 16).

## Factors associated with healthcare waste handling practice

In the bivariate logistic regression analysis; Sex, age group, occupational category, work experience, type of health facility, separate container for HCW, located in appropriate place, leak proof materials used for HCW collection, labelling or marking of HCW container, easy to carry by the handlers, puncture- resistant material for sharps, HCW containers emptied daily or whenever ¾ full, formal or informal separation of HCW takes place, recycling of used plastic materials, HCW handlers wear heavy duty gloves and sturdy shoes, wash both hard heavy duty gloves and hands after handling HCW, means of transportation for HCW and generation of HCW of special concern (cytotoxic) showed statistically significant association with separate storage area for healthcare waste (Table 17).

The backward stepwise multivariate logistic regression analysis has shown that the odds of healthcare waste handling practice was found to increase by 5 times among using puncture resistant material for sharps [AOR = 4.82, 95% CI: (2.32, 10.02)]. The generation of cytotoxic waste had an association with the healthcare waste handling practice. Generation of cytotoxic waste [AOR = 8.37, 95% CI: (3.20, 21.88)] were 8.37 times more likely to health care waste handling practice (Table 17).

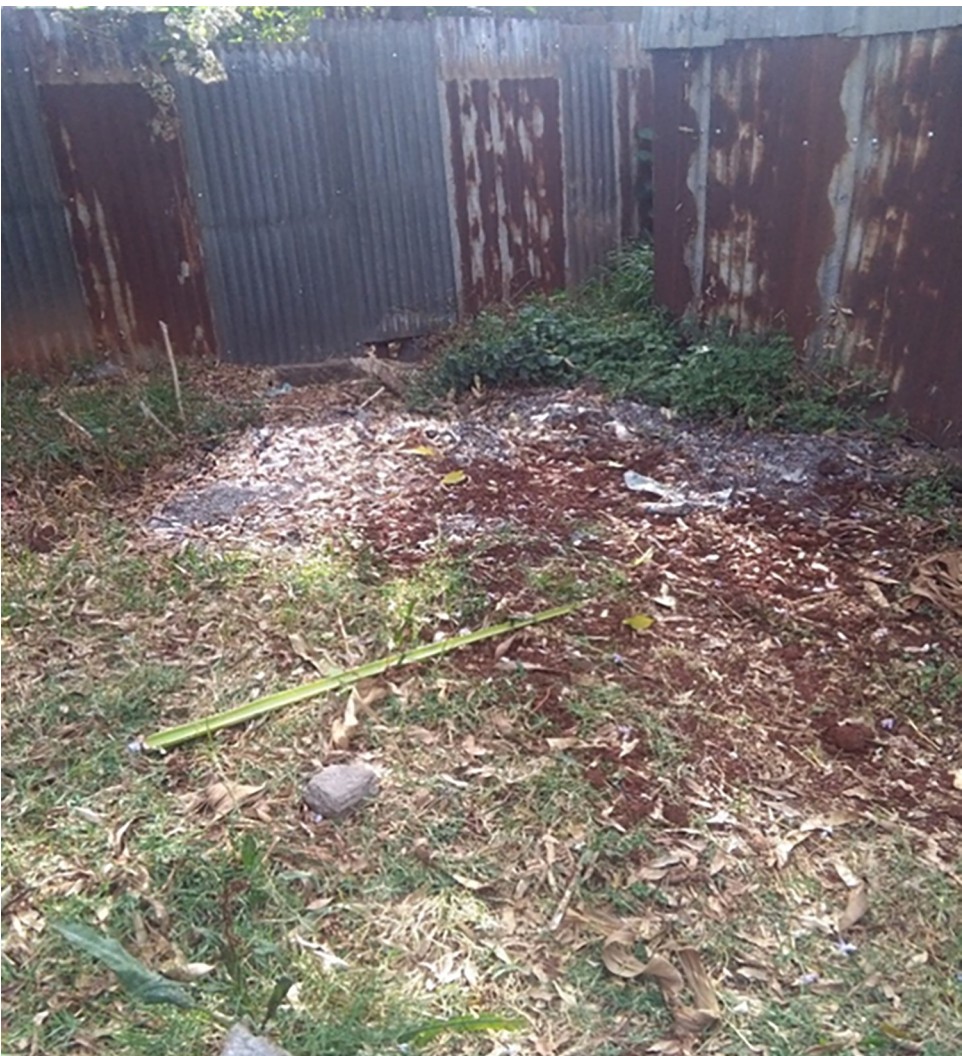

**Fig 9. Ash disposal inside the compound of one of the study health care facilities, Addis Ababa City Administration, February 2018.**

## Discussion

From previous study done in Addis Ababa half of the health centers didn't have separate containers for the collection of hazardous and non-hazardous wastes moreover the labeling of the waste containers didn't see by seven of the study health centers [2]. In this study most respondents, 358 (90.86%) from the health centres and 133 (96.38%) from the hospitals indicated that their facilities had separate containers for hazardous and non-hazardous waste also some respondents 15.48% (61) from the health centres and 29 (21.01%) from the hospitals indicated that the healthcare waste containers were not clearly marked or labelled. Two hundred and forty-one, (67.3%) used the readily existing waste bins for placing of medical waste in South Omo Zone public health facilities [15]. The difference might be in organizing different management structure in the health care facilities.

In this study collection of HCW in the healthcare facilities was not regularly done, 55 (13.96%) from the health centres and 32 (23.19%) from the hospitals indicated that the HCW containers were not located in appropriate areas where they might be needed. The study from

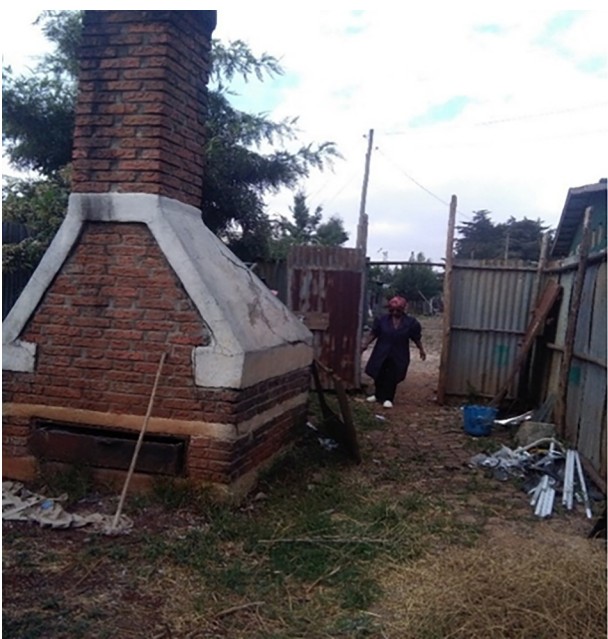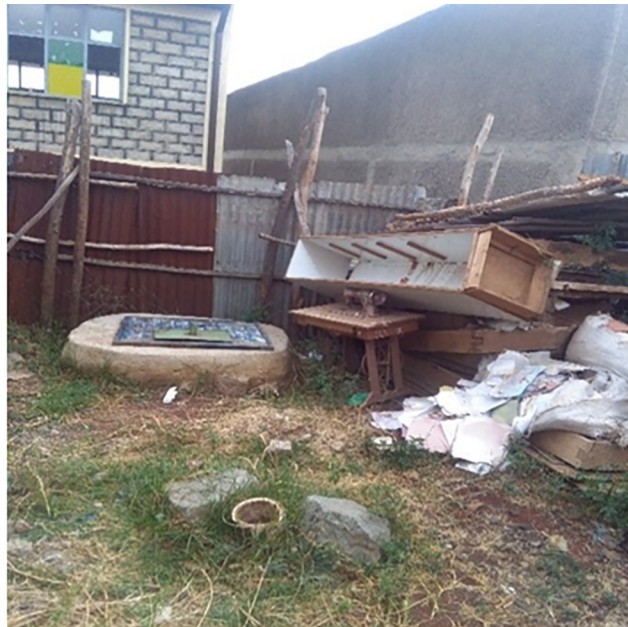

**Fig 10.** Incinerator (a) and placenta pit (b) with no fencing in one of the study health centres, Addis Ababa City Administration, February 2018.

three hospitals of Addis Ababa indicated, HCW materials were collected daily basis while the collection program was irregular in the most hospitals [11]. The reason might be either there is a shortage of containers or negligence by the coordinators.

HCW containers in the health facilities were not made of leak-proof material it was indicated by the respondents, 69 (17.51%) from the health centres and 33 (24.26%) from the hospitals. It is also similar to the study done in Addis Ababa most of the HCW at the hospitals was found to be collected in perforated plastic bins that are intended for use in administrative areas only [11].

The study found that HCW containers were not easy to carry for transportation in the healthcare facilities, it was indicated by the respondents, 64 (16.24%) from the health centres and 39 (28.26%) from the hospitals. Most health centres mainly used buckets followed by pedal bins and trolleys to transport HCW on site while the hospitals used mainly pedal bins and sometimes buckets and trolleys. The study conducted in Addis Ababa private clinics showed 16 (5.8%) of the clinics had trolley/wheelbarrow and 2 (0.7%) of the clinics were linked with the sewerage lines [16]. The overall waste transporting practice was poor in 238 (85.6%) of the clinics [16]. The similarity might be the healthcare facilities consider to transporting HCW be the lower priority to manage.

**Table 16. Collection and off-site disposal of HCW from the study health facilities, Addis Ababa City Administration, February 2018.**

| Collection and off-site disposal of HCW | Health centre (n = 15) | Hospital (n = 3) |
|---|---|---|
| Open burning | 1 | - |
| Incineration | 14 | 3 |
| Municipality | - | 3 |
| Cooperatives | 3 | - |
| No disposal off-site | 12 | - |

**Table 17. Factors associated with HCW handling practice among HCW handlers in the study health facilities, Addis Ababa City Administration, February 2018.**

| Variable | | Separate storage area for HCW | | Crude OR | Adjusted OR |
| --- | --- | --- | --- | --- | --- |
| | | | | No (95% CI) | No (95% CI) |
| | | Yes | No | | |
| Sex | Male | 194 | 21 | 1.00 | |
| | Female | 283 | 34 | 0.901(0.508,1.599) | |
| Age group | 20–35 | 426 | 44 | 2.088(1.015,4.297)* | |
| | 36+ | 51 | 11 | 1 | |
| Occupational category | Doctors, Nurses and Midwives | 267 | 28 | 0.605(0.205,1.788) | |
| | Pharmacist and Laboratory Technology | 95 | 14 | 0.431(0.136, 1.369) | |
| | Ancillary staff | 52 | 9 | 0.367(0.107,1.260) | |
| | Health officer Biomedical engineer, Environmental health and Radiographer | 63 | 4 | 1.00 | |
| Work experience | 1–10 | 444 | 47 | 2.29(1.010,5.246)* | |
| | 11+ | 33 | 8 | 1 | |
| Type of health facility | Health centre | 350 | 44 | 0.69(0.345,1.375) | |
| | Hospital | 127 | 11 | 1.00 | |
| Separate container for HCW | Yes | 448 | 43 | 4.31 (2.053,9.054)*** | |
| | No | 29 | 12 | 1.00 | |
| Located in appropriate place | Yes | 408 | 37 | 2.88 (1.550,5.338)*** | |
| | No | 69 | 18 | 1.00 | |
| Leak proof materials used for HCW collection | Yes | 393 | 37 | 2.28(1.236,4.191)** | |
| | No | 84 | 18 | 1.00 | |
| Labelling or marking of HCW container | Yes | 405 | 37 | 2.74(1.477,5.069)*** | |
| | No | 72 | 18 | 1.00 | |
| Containers easy to carry by the handlers | Yes | 398 | 31 | 3.90(2.173,7.001)*** | |
| | No | 79 | 24 | 1.00 | |
| Puncture- resistant material for sharps | Yes | 432 | 30 | 8.0(4.333,14.770)*** | 4.824 (2.324,10.015)*** |
| | No | 45 | 25 | 1.00 | |
| HCW containers emptied daily or whenever ¾ full | Yes | 389 | 33 | 2.95(1.639,5.300)*** | |
| | No | 88 | 22 | 1.00 | |
| Formal or informal separation of HCW takes place | Yes | 364 | 31 | 2.49(1.406,4.424)** | |
| | No | 113 | 24 | 1.00 | |
| Recycling of used plastic materials | Yes | 226 | 10 | 4.05(1.995,8.228)*** | |
| | No | 251 | 45 | 1.00 | |
| HCW handlers wear heavy duty gloves and sturdy shoes | Yes | 328 | 29 | 1.97(1.123,3.468) * | |
| | No | 149 | 26 | 1.00 | |
| Wash both heavy-duty gloves and hands after handling HCW | Yes | 372 | 31 | 2.74(1.543,4.876)*** | |
| | No | 105 | 24 | 1.00 | |

(*Continued*)

**Table 17.** (Continued)

| Variable | | Separate storage area for HCW | | Crude OR No (95% CI) | Adjusted OR No (95% CI) |
|---|---|---|---|---|---|
| | | Yes | No | | |
| Means of transportation for HCW | Cart | 136 | 8 | | |
| | Open bucket | 305 | 43 | 2.00(0.569,7.035) | |
| | Pedal bin | 34 | 4 | 0.834(0.282,2.467) | |
| | Trolley | 2 | 0 | 1.00 | |
| | | | | | |
| Generation of HCW of special concern | | | | | |
| Cytotoxic | Yes | 213 | 5 | 8.07(3.16,20.590)*** | 8.37 (3.202,21.875)*** |
| | No | 264 | 50 | 1.00 | |
| Pathological | Yes | 324 | 33 | 1.42(0.796,2.503) | |
| | No | 153 | 22 | 1.00 | |
| Reagent | Yes | 294 | 28 | 1.55(0.885,2.712) | |
| | No | 183 | 27 | 1.00 | |
| Outdated pharmaceutical | Yes | 302 | 34 | 1.07(0.600,1.894) | |
| | No | 175 | 21 | 1.00 | |
| Radioactive | Yes | 87 | 7 | 1.53(0.669,3.495) | |
| | No | 390 | 48 | 1.00 | |

*P< 0.05

**P< 0.01

***P< 0.001

In this study 310(76.65%) from health centres and 93(67.39%) from hospitals indicated that healthcare waste handlers washed their hands after handling wastes. Study done in Addis Ababa government hospitals, 57.6% professionals followed standard precaution practice after any direct contact to patients and their disposals [17]. This indicates most of the workers has a good practice to standard precaution practices.

Off-site disposal of HCW implemented in the healthcare facilities was assured by the respondents, 14 (20%) from the health centres and 7 (20.60%) from the hospitals and most of them indicated that the municipality collected the HCW for off-site disposal. Similarly, the study conducted in Addis Ababa hospitals showed that dispose their waste at off-site, the untreated hospital waste materials in the central storage area were finally loaded onto vehicles and transported to "koshe" unsanitary landfill site for open dumping [11]. This might be the healthcare facilities has a problem to treatment or disinfection of HCW which pose infection to human and the environment.

In this study application of operational standards and guidelines for HCW management in the healthcare facilities also limited, it was indicated by respondent managers from the health centres, 23 (32.86%) and from hospital 13 (38.23%) indicated there was no current operational standard for HCW management. In previous study conducted in Addis Ababa six out of ten studied health centers, Standard Operational Procedures, as well as any applicable local or regional guidelines about healthcare waste management were not found [2]. But another study done in Bahir Dar private and public hospitals 161(83.9%) and 179(79.2%) respectively indicated healthcare workers responded as there were rules and regulations regarding HCW

management in the health facilities [18] also study done in South Omo Zone public health facilities indicated 41.3% of respondents apply medical waste management guidelines and policy to manage health-care waste correctly [15]. The reason might be either preparation of HCW management guideline by health facilities or the policy makers do consider HCW as an issue of priority.

In this study indicated most healthcare facilities had no HCW management committees 13 (18.57%) and 11 (32.35%) from the health centres and hospitals respectively also study done in Bahir Dar 59(30.7%) of private hospitals and 86(38.1%) of public hospitals health care workers indicated had no healthcare waste management committee in the health facilities [18].

Incineration was the most common method of treatment for HCW in studied healthcare facilities in Addis Ababa. Similar studies in Belo Horizonte, Brazil showed 60% of HCW treatment technology goes directly to incineration [19]. There is no centralized incineration for all HCFs in Addis Ababa and surrounding regions to destroy pharmaceutical wastes. Most of the study HCFs had incinerators on the premises; only a few incinerators were located downwind from the main service area burn all hazardous and non-hazardous waste together. Most incinerators had sufficient air inlets on the side in most cases ash from the incinerators was disposed of inside the compound. Many of the incinerators were not surrounded by a fence or wall to limit access to scavengers. The finding in line with other hospitals and private clinics study in Addis Ababa showed the main HCW disposal mechanism was incineration, incinerators incinerating all the solid HCWs together and used low combustion, single chamber, brick incinerators, and barrels in clinics incinerator as a treatment/final waste disposal method [11, 16]. A systematic review done in Ethiopia waste treatment and disposal practice indicated low combustion incinerator was used to treat all the HCW types [20]. This might be due to lack of proper way of quantifying the types of waste management utility supply, poor financial allocation and rules and regulations.

## Conclusion

Healthcare waste management system had been given very little attention in all health centers and hospitals. Pretreatment of infectious solid waste and liquid waste must be practiced before disposing helps to minimize the transmission of most pathogens to human and environment. Intervention measures are important point to fill the gap in knowledge, practice and attitude should be supported by training on healthcare waste management for waste handlers and managers bring greatest change on practice and management of HCWs. Healthcare facilities collaborate with private and non-government organization as partners or other stake holders also important strengthening Public Private Partnership is very important. The presence of applicable national, regional and local guidelines for HCW management practice is helpful for all healthcare facilities to guide all aspects in HCW management. The findings of the study should contribute to the achievement of the United Nations [21] sustainable development goals (SDGs) for 2016–2030, which are aimed at bringing about a sustainable world and protecting the planet.

## Limitation of the study

This study has the following limitation: the study conducted was cross-sectional and couldn't identify causality. The study was conducted in public healthcare facilities healthcare waste handlers and managers and couldn't represent healthcare waste handlers and managers outside the public healthcare facilities (private HCF). The study is conducted on healthcare waste management issues other studies should also be conducted the generation rate is very important.

## Acknowledgments

First, I would like to express my deepest gratitude to Professor Bethabile Lovely Dolamo for her unreserved support throughout the study period. I sincerely thank University of South Africa, Kotebe Education University Menelik II Medical and Health Science College, Addis Ababa City Government Health Bureau, head of the study health centers and hospital case teams and managers for their unreserved cooperation during data collection time. My deepest gratitude also goes to all data collectors and supervisor for their commitment during data collection. I would like to thank my beloved wife Alemnesh Mude, daughters Bezawit Menelik and Hermela Menelik for their patience during the study period.

## Author Contributions

**Methodology:** Bethabile Lovely Dolamo.

**Writing – original draft:** Menelik Legesse Tadesse.

**Writing – review & editing:** Menelik Legesse Tadesse, Bethabile Lovely Dolamo.

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
