## [Decision Letter · Decision Letter 0]

27 Jun 2022

PONE-D-21-40419

Assessment of healthcare waste management practices and associated factors in Addis Ababa City Administration Public Health Facilities.

PLOS ONE

Dear Dr. Tadesee,

Thank you for submitting your manuscript to PLOS ONE. After careful consideration, we feel that it has merit but does not fully meet PLOS ONE’s publication criteria as it currently stands. Therefore, we invite you to submit a revised version of the manuscript that addresses the points raised during the review process.

We look forward to receiving your revised manuscript.

Kind regards,

Balasubramani Ravindran, Ph.D

Academic Editor

PLOS ONE

2. (1) Please provide additional details regarding participant consent. In the ethics statement in the Methods and online submission information, please ensure that you have specified what type you obtained (for instance, written or verbal, and if verbal, how it was documented and witnessed). If your study included minors, state whether you obtained consent from parents or guardians. If the need for consent was waived by the ethics committee, please include this information.

Reviewers' comments:

Reviewer's Responses to Questions

**Comments to the Author**

1. Is the manuscript technically sound, and do the data support the conclusions?

Reviewer #1: Partly

Reviewer #2: Partly

2. Has the statistical analysis been performed appropriately and rigorously? 

Reviewer #1: Yes

Reviewer #2: Yes

3. Have the authors made all data underlying the findings in their manuscript fully available?

Reviewer #1: Yes

Reviewer #2: Yes

4. Is the manuscript presented in an intelligible fashion and written in standard English?

Reviewer #1: Yes

Reviewer #2: No

5. Review Comments to the Author

Reviewer #1: This manuscript tries to assess the healthcare waste management practices and associated factors in Addis Ababa City Administration Public Health Facilities. I believe it merits publication after shortening and inclusion of the following comments

Abstract.

1. The data are not presented in the abstract and it is more of a qualitative description

2. conclusion is missing and the authors should follow the guidelines of the journal

Introduction

1. Background to the research problem section should be combined with background section into a single section

2. Previous research conducted in the area should be addressed in the introduction section and make comparisons using results from the following articles

1. Assessment of the health care waste generation rates and its management system in hospitals of Addis Ababa, Ethiopia, 2011,. https://doi.org/10.1186/1471-2458-13-28

2. Biomedical waste management practices and associated factors among health care workers in the era of the covid-19 pandemic at metropolitan city private hospitals, Amhara region, Ethiopia, 2020. https://doi.org/10.1371/journal.pone.0266037

3. Healthcare Waste Management Practices and Associated Factors in Private Clinics in Addis Ababa, Ethiopia. https://doi.org/10.1177/11786302211073383

4.(Pichtel 2014:549) referencing issue

Materials methods

1. Inclusion and exclusion criteria is missing in the materials and methods. The authors should address this.

2. The study period is not indicated in the current version of the manuscript

Result section

1. This section is too lengthy and I would suggest to be presented through several sub-heading as

• types of waste generated

• collection and transportation, facility

• Health care waste separation, etc

2. In the sub heading ‘Healthcare waste management’

I doubt the clinics shall contain 105 mangers and please reflect on this

3. ‘Of the respondents’, repeated several time and it is unpleasant to the readers

Discussion

This section is simply a repetition of results obtained in the study. It does not attempt to discuss the results, nor does it contain any references. It's hard to say it's a discussion

Shorten your conclusion and make them more quantitative.

Reviewer #2: Points needed to be addressed1. Are the authors assessing the knowledge of the healthcare professionals or workers on the waste management facilities present at their respective institution? If yes it makes sense for them to know or not know. If not how do you explain them replying NO to a clearly visible practice?2. Table 4 represents HCW handlers management practice at the study health facitilities, again here it does not mention the knowledge of the participants, but of the practice being carried out which has a fixed response by observation. Same goes for table 5 and 6.The language needs to be addressed to suit the journals standardAuthors have not provided any discussion based on other studies. Does that make it a novel work on this? Kindly clarify it. As all studies should have a discussion section that discuses the results based on findings of other studies. Authors have clearly not provided any such which is indicated by the lack of references used. This is not an acceptable way of submission.I advise a thorough re check of the manuscript to address the given concerns.

6. PLOS authors have the option to publish the peer review history of their article (what does this mean?). If published, this will include your full peer review and any attached files.

Reviewer #1: No

Reviewer #2: No

---

## [Author Response · Author response to Decision Letter 0]

2 Aug 2022

I would like to thank all the reviewers for your constructive comments. It is my pleasure to see your response soon.

---

## [Decision Letter · Decision Letter 1]

29 Aug 2022

PONE-D-21-40419R1Assessment of healthcare waste management practices and associated factors in Addis Ababa City Administration Public Health Facilities.PLOS ONE

Dear Dr. Tadesse,

Thank you for submitting your manuscript to PLOS ONE. After careful consideration, we feel that it has merit but does not fully meet PLOS ONE’s publication criteria as it currently stands. Therefore, we invite you to submit a revised version of the manuscript that addresses the points raised during the review process.

We look forward to receiving your revised manuscript.

Kind regards,

Nor Adilla Rashidi, Ph.D.

Academic Editor

PLOS ONE

Reviewers' comments:

Reviewer's Responses to Questions

**Comments to the Author**

1. If the authors have adequately addressed your comments raised in a previous round of review and you feel that this manuscript is now acceptable for publication, you may indicate that here to bypass the “Comments to the Author” section, enter your conflict of interest statement in the “Confidential to Editor” section, and submit your "Accept" recommendation.

Reviewer #1: (No Response)

Reviewer #3: All comments have been addressed

2. Is the manuscript technically sound, and do the data support the conclusions?

Reviewer #1: (No Response)

Reviewer #3: Yes

3. Has the statistical analysis been performed appropriately and rigorously? 

Reviewer #1: (No Response)

Reviewer #3: N/A

4. Have the authors made all data underlying the findings in their manuscript fully available?

Reviewer #1: (No Response)

Reviewer #3: Yes

5. Is the manuscript presented in an intelligible fashion and written in standard English?

Reviewer #1: (No Response)

Reviewer #3: Yes

6. Review Comments to the Author

Reviewer #1: The authors have improved the manuscript significantly, and I believe it is worth publishing after incorporation few published articles to the discussion section. The authors used three references to summarize their findings. Does that mean that most of their findings provide first-hand information? The section merely presents the results without discussing it. Nevertheless, I recommend enriching the discussion with

1. https://doi.org/10.1007/s11356-022-22113-w

2. https://doi.org/10.1155/2020/7837564

3. http://dx.doi.org/10.1136/bmjopen-2019-030784

4. https://doi.org/10.1186/s13104-019-4316-y

5. https://doi.org/10.2147/JHL.S300729

6. https://doi.org/10.1371/journal.pone.0266037

7. And many more articles

Reviewer #3: As the comments are long, please see the comments in the separate attached file.

In general, the paper has good information.

7. PLOS authors have the option to publish the peer review history of their article (what does this mean?). If published, this will include your full peer review and any attached files.

Reviewer #1: No

Reviewer #3: No

---

## [Author Response · Author response to Decision Letter 1]

9 Oct 2022

I would like to thank all the reviewers for your constructive comments. I am happy for all comments.

---

## [Decision Letter · Decision Letter 2]

24 Oct 2022

Assessment of healthcare waste management practices and associated factors in Addis Ababa City Administration Public Health Facilities.

PONE-D-21-40419R2

Dear Dr. Tadesse,

We’re pleased to inform you that your manuscript has been judged scientifically suitable for publication and will be formally accepted for publication once it meets all outstanding technical requirements.

Kind regards,

Nor Adilla Rashidi, Ph.D.

Academic Editor

PLOS ONE

Additional Editor Comments (optional):

Reviewers' comments:

Reviewer's Responses to Questions

**Comments to the Author**

1. If the authors have adequately addressed your comments raised in a previous round of review and you feel that this manuscript is now acceptable for publication, you may indicate that here to bypass the “Comments to the Author” section, enter your conflict of interest statement in the “Confidential to Editor” section, and submit your "Accept" recommendation.

Reviewer #1: (No Response)

Reviewer #3: All comments have been addressed

2. Is the manuscript technically sound, and do the data support the conclusions?

Reviewer #1: (No Response)

Reviewer #3: Yes

3. Has the statistical analysis been performed appropriately and rigorously? 

Reviewer #1: (No Response)

Reviewer #3: Yes

4. Have the authors made all data underlying the findings in their manuscript fully available?

Reviewer #1: (No Response)

Reviewer #3: Yes

5. Is the manuscript presented in an intelligible fashion and written in standard English?

Reviewer #1: (No Response)

Reviewer #3: Yes

6. Review Comments to the Author

Reviewer #1: (No Response)

Reviewer #3: (No Response)

7. PLOS authors have the option to publish the peer review history of their article (what does this mean?). If published, this will include your full peer review and any attached files.

Reviewer #1: No

Reviewer #3: No

---

## [Editor Report · Acceptance letter]

26 Oct 2022

PONE-D-21-40419R2 

Assessment of healthcare waste management practices and associated factors in Addis Ababa City Administration Public Health Facilities. 

Dear Dr. Tadesse:

I'm pleased to inform you that your manuscript has been deemed suitable for publication in PLOS ONE. Congratulations! Your manuscript is now with our production department. 

Kind regards, 

on behalf of

Dr. Nor Adilla Rashidi 

Academic Editor

PLOS ONE